# Fast transient networks in spontaneous human brain activity

**Adam P Baker[1,8]\***, **Matthew J Brookes[2]**, **Iead A Rezek[3†]**, **Stephen M Smith[4]**, **Timothy Behrens[5,6]**, **Penny J Probert Smith[7]**, **Mark Woolrich[4,1]\***

[1]Oxford Centre for Human Brain Activity, University of Oxford, Oxford, United Kingdom; [2]Sir Peter Mansfield Magnetic Resonance Centre, School of Physics and Astronomy, University of Nottingham, Nottingham, United Kingdom; [3]Department of Engineering Science, University of Oxford, Oxford, United Kingdom; [4]Oxford Centre for Functional MRI of the Brain, Nuffield Department of Clinical Neuroscience, University of Oxford, Oxford, United Kingdom; [5]Oxford Centre for Functional MRI of the Brain, Nuffield Department of Clinical Neuroscience, Oxford University, Oxford, United Kingdom; [6]Wellcome Trust Centre for Neuroimaging, University College London, London, United Kingdom; [7]Department of Engineering Science, University of Oxford, Oxford, United Kingdom; [8]Centre for Doctoral Training in Healthcare Innovation, Institute of Biomedical Engineering, Department of Engineering Science, University of Oxford, Oxford, United Kingdom

**Abstract** To provide an effective substrate for cognitive processes, functional brain networks should be able to reorganize and coordinate on a sub-second temporal scale. We used magnetoencephalography recordings of spontaneous activity to characterize whole-brain functional connectivity dynamics at high temporal resolution. Using a novel approach that identifies the points in time at which unique patterns of activity recur, we reveal transient (100–200 ms) brain states with spatial topographies similar to those of well-known resting state networks. By assessing temporal changes in the occurrence of these states, we demonstrate that within-network functional connectivity is underpinned by coordinated neuronal dynamics that fluctuate much more rapidly than has previously been shown. We further evaluate cross-network interactions, and show that anticorrelation between the default mode network and parietal regions of the dorsal attention network is consistent with an inability of the system to transition directly between two transient brain states.

**\*For correspondence:** adam.baker@ohba.ox.ac.uk (APB); mark.woolrich@ohba.ox.ac.uk (MW)

**Present address:** †The Schlumberger Gould Research Center, Cambridge, United Kingdom

**Reviewing editor**: Jody C Culham, University of Western Ontario, Canada

## Introduction

The presence of large-scale distributed networks of temporally correlated spontaneous activity is a well-established phenomenon in neuroimaging (*Biswal et al., 1995*; *Fox and Raichle, 2007*; *Raichle et al., 2001*). These so-called resting state networks (RSNs) have been consistently identified across different subjects in the absence of any explicit task from covariations in the blood oxygenation level dependent (BOLD) signal, as measured by functional magnetic resonance imaging (fMRI) (*Beckmann et al., 2005*; *Damoiseaux et al., 2006*; *Smith et al., 2009*). These networks are known to have functional relevance and clinical significance (*Greicius et al., 2004*; *Filippini et al., 2009*).

However, an important limitation in our understanding of spontaneous activity is how the low frequency fluctuations typically associated with RSNs are related to the much faster timescales of cognitive and sensory processing (*Heeger and Ress, 2002*; *Deco and Corbetta, 2011*; *Raichle, 2011*; *Siegel et al., 2012*). This is in part due to the fact that the indirect hemodynamic response measured via the BOLD signal precludes studying the rich temporal dynamics of the underlying electrophysiological activity (*Logothetis, 2008*). In contrast, non-invasive electrophysiological recordings such as

**eLife digest** When subjects lie motionless inside scanners without any particular task to perform, their brains show stereotyped patterns of activity across regions known as resting state networks. Each network consists of areas with a common function, such as the 'motor' network or the 'visual' network. The role of resting state networks is unclear, but these spontaneous activity patterns are altered in disorders including autism, schizophrenia, and Alzheimer's disease.

One puzzling feature of resting state networks is that they seem to last for relatively long times. However, the majority of studies into resting state networks have used fMRI brain scans, in which changes in the level of oxygen in the blood are used as a proxy for the activity of a given brain region. Since changes in blood oxygen occur relatively slowly, the ability of fMRI to detect rapid changes in activity is limited: it is thus possible that the long-lived nature of resting state networks is an artefact of the use of fMRI.

Now, Baker et al. have used a different type of brain scan known as an MEG scan to show that the activity of resting state networks is shorter lived than previously thought. MEG scanners measure changes in the magnetic fields generated by electrical currents in the brain, which means that they can detect alterations in brain activity much more rapidly than fMRI.

MEG recordings from the brains of nine healthy subjects revealed that individual resting state networks were typically stable for only 100 ms to 200 ms. Moreover, transitions between different networks did not occur randomly; instead, certain networks were much more likely to become active after others. The work of Baker et al. suggests that the resting brain is constantly changing between different patterns of activity, which enables it to respond quickly to any given situation.

magnetoencephalography (MEG) and electroencephalography (EEG) provide a direct measure of neuronal activity at high temporal resolution. Recent studies using these modalities have revealed that these networks have an electrophysiological basis (*Laufs et al., 2003*; *He et al., 2008*; *Jann et al., 2010*; *Liu et al., 2010*; *Brookes et al., 2011*), and are underpinned by much richer spatiotemporal dynamics that may be better characterized using time-varying measures of interactions (*Brookes et al., 2014*; *Chang et al., 2013*; *Hutchison et al., 2013*; *de Pasquale et al., 2010*, *2012*). These studies have shown evidence that functional connectivity within whole brain networks exhibit temporal variability on a time scale of seconds to tens of seconds. However, to provide an effective substrate for cognitive processes, functional networks should be able to rapidly reorganize and coordinate on a sub-second temporal scale (*Bressler and Tognoli, 2006*).

Here, we present a study that identifies transient networks of brain activity, with no prior assumptions on the brain areas or time scales involved. This uses a distinct methodology based on a hidden Markov model (HMM), which infers a number of discrete brain states that recur at different points in time. Each inferred state corresponds to a unique pattern of whole-brain spontaneous activity, which is modeled by a multivariate normal distribution and a state time course indicating the points in time at which that state is active. These two outputs are shown schematically in *Figure 1*, and allow us to describe both the spatial and temporal characteristics of each inferred state.

By applying this methodology to band-limited amplitude envelopes of source reconstructed MEG data, we show that the HMM can independently identify brain states in MEG data that correspond to established RSNs, and which fluctuate at time scales two orders of magnitude faster than has previously been shown. We further assess cross-network interactions, and show that the antagonistic behavior between the default mode network (DMN) and parietal regions of the dorsal attention network (DAN) is consistent with an inability of the system to transition directly between two of these states.

## Results

We recorded 10 min of resting state MEG data from nine healthy subjects. We applied beamforming and frequency filtering to derive spatially and spectrally resolved estimates of neuronal activity. Data across multiple subjects were temporally concatenated resulting in a single group data matrix from which an HMM with 8 states was inferred (see 'Materials and methods'). The inferred HMM states represent unique spatiotemporal patterns of activity (band-limited amplitude fluctuations) that repeat at different points in time (*Rezek and Roberts, 2005*).

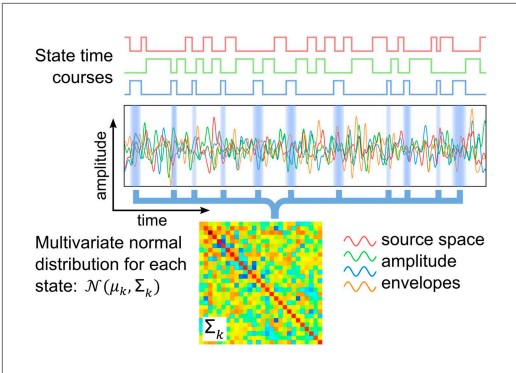

**Figure 1**. Schematic of the HMM outputs. An HMM with $K$ states is inferred from band-limited amplitude envelopes of source reconstructed MEG data. Each state is characterized by a multivariate normal distribution (defined by means $\mu_k$ and covariance matrix $\Sigma_k$) and a state time course, which is a binary sequence that indicates the points in time at which the state is active.

## Large-scale transient networks of spontaneous oscillatory activity

State specific changes in oscillatory amplitude revealed spatial patterns of activity with good similarity to several well-known networks, previously associated with brain wide correlations at slow (<0.1 Hz) timescales (*Figure 2*). Each map represents the partial correlation between the state time courses and the group-concatenated amplitude envelope at each voxel. Accordingly, state specific increases and decreases in amplitude are represented by red/yellow and blue colors respectively. State 1 shows increased activity in nodes of the default mode network (DMN) including left and right inferior parietal lobule, medial frontal gyrus and medial temporal lobe (but notably, not in the posterior cingulate/precuneus). States 2–6 show increased activity in the visual cortex (states 2 and 6), the sensorimotor network (state 3), and the left and right lateralized temporal lobes (states 4 and 5). States 7–8 show decreases in activity (blue in *Figure 2*) in parietal regions including the intraparietal sulcus (IPS; state 7), and visual cortex (state 8).

The temporal properties of each state were characterized from the state time courses, which indicate the points in time at which each state is active. By inspection it is evident that the states are short lived (*Figure 3A*). The temporal characteristics of each state may be quantified in terms of their fractional occupancy (fraction of the total time spent in a state; *Figure 3B*), life times (time spent in a state before making a transition; *Figure 3C*), and interval lengths (time between consecutive state visits; *Figure 3D*). Average life times are between 100 ms and 200 ms. These life times are markedly shorter in duration than the time scales typically associated with resting state networks, which have previously been shown to be dominated by frequencies below 0.1 Hz (*Cordes et al., 2001*). We also characterized variations in the rate at which these states are visited by computing the fractional occupancy within a 10-s sliding window. These fractional occupancy time courses reveal slower temporal changes in the occurrence of the HMM states. These time courses are shown for all subjects in *Figure 3E*. It is clear that each state was represented in all subjects.

## Large-scale networks of oscillatory activity are underpinned by rapid fluctuations

We have shown that whole brain spontaneous activity may be broken down into a set of distinct connectivity patterns that appear to be stable for periods of 100–200 ms. To confirm that these brain states are consistent with coordinated fluctuations at these rapid time scales, we performed a follow up analysis using the inferred state time courses. We reasoned that if there were coordinated fluctuations at the fastest time scales in the state time course, then low pass filtered versions of the state time courses should do worse at explaining fluctuations in the data. Different low pass filtered versions of the state time courses were obtained by computing fractional occupancy time courses using a range of time windows from 0.1 to 8 s. Note that for the shortest time windows, the fractional occupancy time course approximates the state time course. These fractional occupancy time courses were then separately regressed onto the amplitude envelope time course from a representative voxel of the corresponding brain state (the voxel that most correlated with the state time course). This analysis reveals a peak in correlation for window widths between 200 ms and 400 ms, demonstrating that when using the fractional occupancy the fastest time scales at which we can detect fluctuations in the amplitude envelopes are slower than those suggested by the HMM state life-times (~100 ms), but still much faster than has been previously shown (*Figure 4*).

As a control, we repeated the HMM inference after first removing any potential high frequency network interactions from the data. The group-concatenated amplitude envelopes were low pass filtered below 0.5 Hz to remove any higher frequency dynamics and an 8 state HMM was inferred

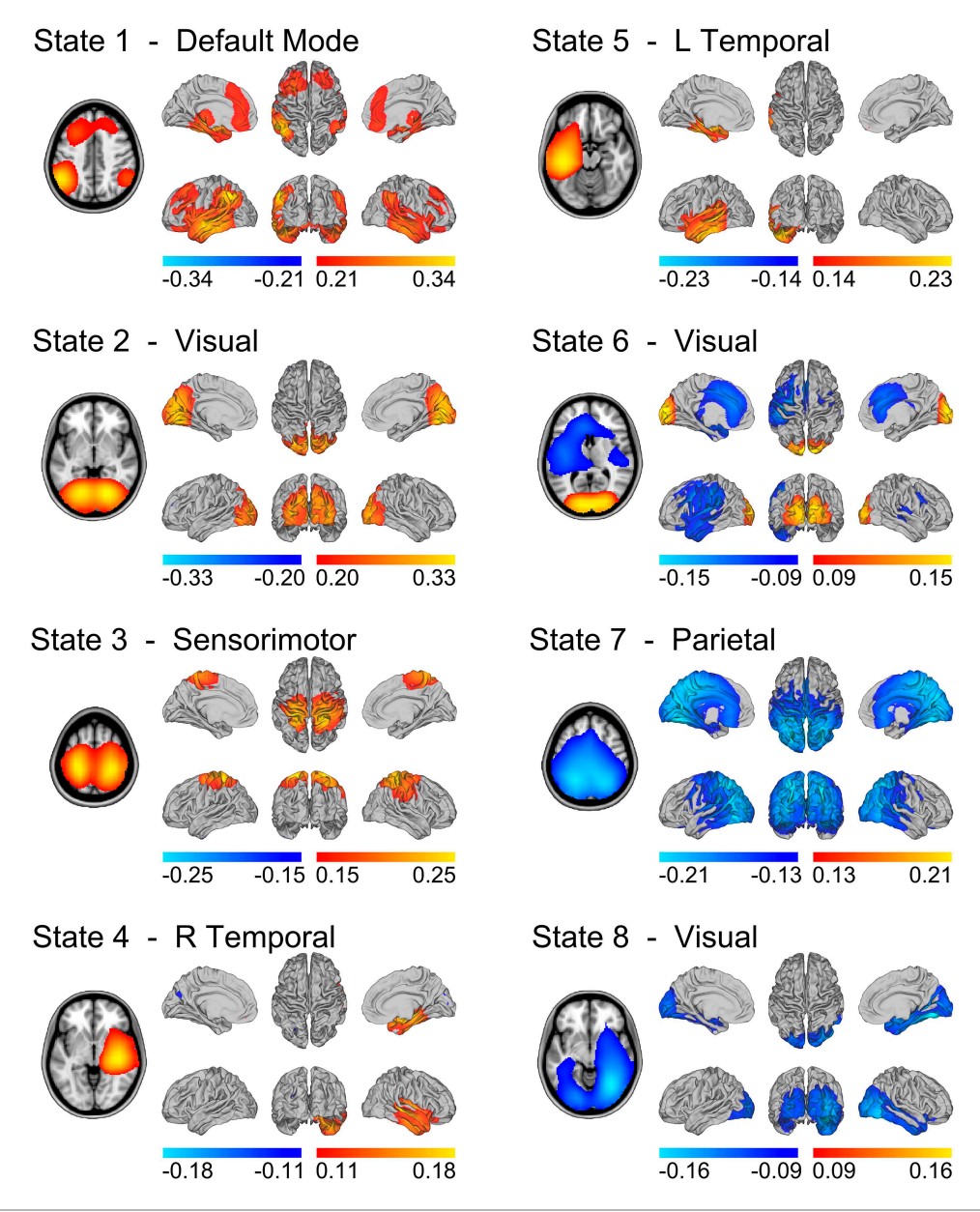

**Figure 2**. State specific changes in band-limited amplitude. An 8 state HMM was inferred from temporally concatenated band-limited amplitude time courses (concatenated over nine subjects, 10 min each). The volumes and surface renderings show the partial correlation of each state time course with the envelope data at each voxel. The correlation values have been thresholded between 60% and 100% of the maximum correlation for each state and the color maps represent these ranges. Red/yellow and blue colors indicate positive and negative correlations respectively. See also *Figure 2—figure supplement 1* for equivalent results from HMMs inferred with 4–14 states.

The following figure supplements are available for figure 2:

**Figure supplement 1**. Maximum intensity projection maps showing the partial correlation computed between each state time course and the envelope data for a k state HMM for k = 4 to k = 14.

**Figure supplement 2**. Spatial maps of five of the inferred states alongside a matched RSN derived from application of spatial ICA to resting state fMRI data (*Smith et al., 2009*).

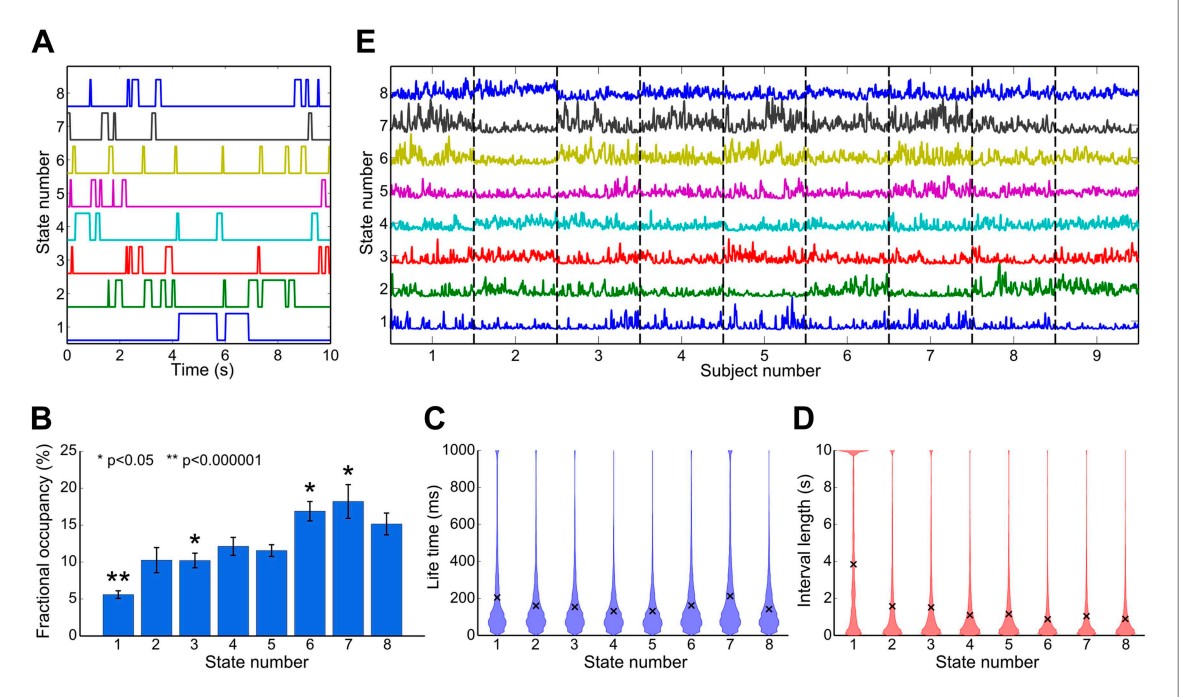

**Figure 3**. Temporal characteristics of the HMM states. (**A**) State time courses showing the most likely state at each time point for the first 10 s of data. (**B**) Fractional occupancy for each inferred state showing the mean and s.e.m. over subjects. The asterisks denote that the fractional occupancy of a state differs significantly from the other states. (**C**) Life times, and (**D**) interval lengths for each inferred state. The filled areas in (**C**) and (**D**) represent the distribution of values and the black crosses show the mean. (**E**) Fractional occupancy of each state as a function of time over all subjects, derived by averaging each state time course within a 10-s sliding window (75% overlap between adjacent windows). See also *Figure 3—figure supplement 1* for a description of how these statistics vary when the HMM is inferred with 4–14 states.

The following figure supplements are available for figure 3:

**Figure supplement 1**. Effect of number of states on (A) model evidence, approximated by the negative of the free energy, (B) minimum fractional occupancy and (C) mean life time, computed over all inferred states and 50 realizations of each HMM inference.

from these filtered envelopes. Despite removing the faster envelope fluctuations, a number of states were inferred with similar spatial topographies as the original HMM (*Figure 4—figure supplement 1A,B*). The life times of these states were longer, at around 1 s, reflecting the slower time scales of these low pass filtered signals. Next, to test whether simply introducing high frequency noise could result in shorter life times, band-limited Gaussian noise (low pass filtered below 10 Hz, reflecting the spectral content of the original envelopes, and therefore with the same non-independence properties between time points), were added to the envelopes prior to inferring the HMM. As before, states with spatial topographies similar to the original HMM states were identified (*Figure 4—figure supplement 1C*). However, the life times of these states were now much shorter, at around 0.3 s.

As with the real data, we then fitted a GLM with a single regressor that corresponded to the fractional occupancy time course computed using a range of time windows from 0.1 s to 8 s. When applied to the low pass filtered dataset (without added Gaussian noise; *Figure 4—figure supplement 1B*), this analysis reveals a peak in correlation for a window width of ~1 s, consistent with the longer state life times. However, for the low pass filtered data set with added Gaussian noise (*Figure 4—figure supplement 1C*), while the state life times were reduced to ~100 ms, the peak in correlation remains at a window width of ~1 s. This demonstrates that in this control there are no detectable within-network amplitude fluctuations faster than 1 s, which is consistent with the applied 0.5 Hz low pass filtering. This is in stark contrast to the real data (*Figure 4*), where the analysis reveals a peak in correlation for window widths between 200 ms and 400 ms.

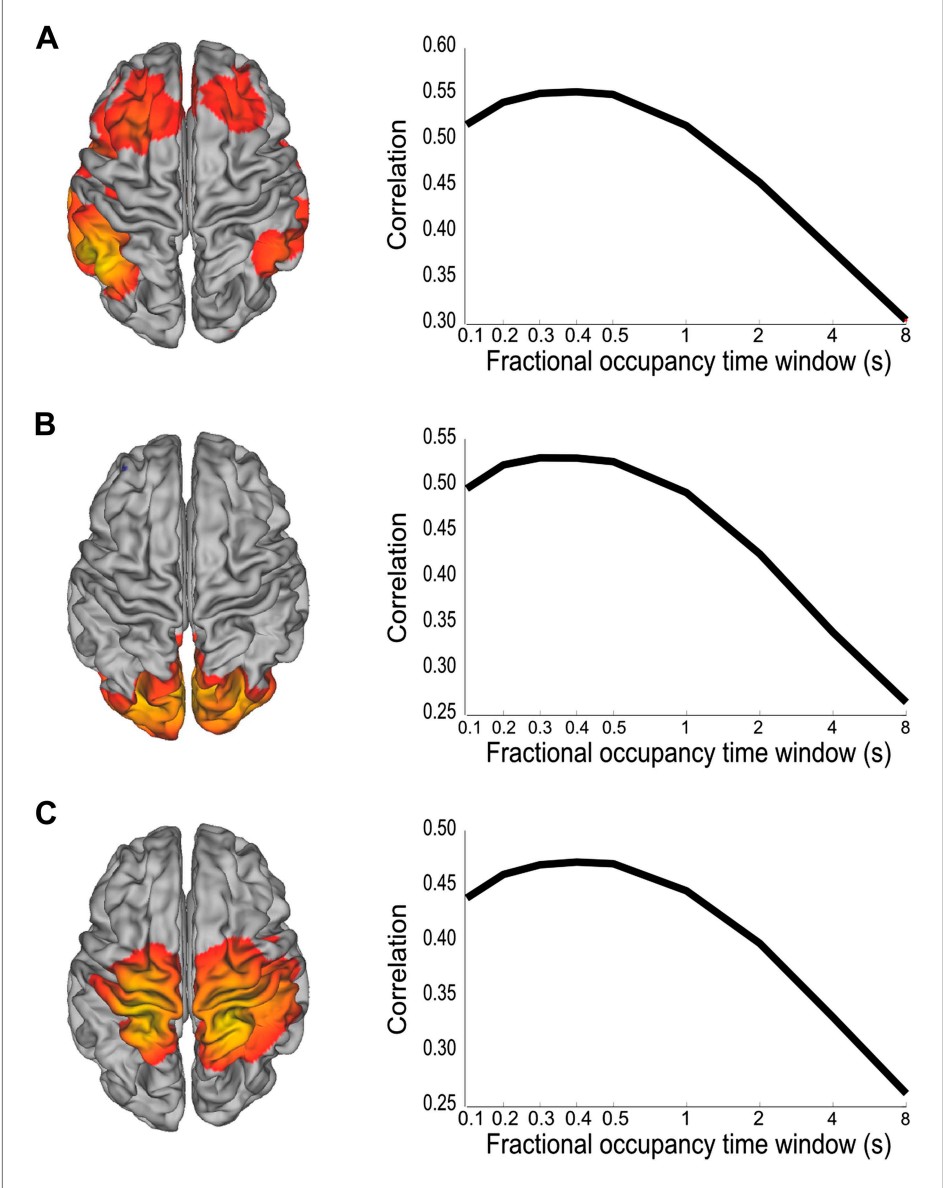

**Figure 4**. Analysis of the time scales that best reflect within-network envelope fluctuations. Partial correlation maps and fractional occupancy time window dependency are shown for (**A**) DMN, (**B**) visual network and (**C**) sensorimotor network. The fractional occupancy time window dependency was computed by fitting a GLM to the amplitude envelope at the voxel with highest correlation with the state time course with the fractional occupancy (computed within different time windows) as a single regressor. See also *Figure 4—figure supplement 1* for a control analysis with surrogate data.

The following figure supplements are available for figure 4:

**Figure supplement 1**. Control analysis of the time scales that best reflect within-network envelope fluctuations.

## State transitions reveal cross-network interactions

Networks of whole brain spontaneous activity may be characterized not only in terms of their within-network activity, but also in terms of cross-network interactions. To assess the relationship between different functional networks in the context of the HMM, we examined the relationship between the points in time at which different states are active. An important result from fMRI studies is the observation that the DMN exhibits anticorrelation with networks associated with attention demanding tasks,

such as the dorsal attention network (DAN) (*Fox et al., 2005*; *Smith et al., 2009*). Of particular interest therefore is the relationship between state 1, which represents the DMN, and state 7, which we postulate may represent parietal regions of the DAN. While the correspondence between BOLD and electrophysiological activity has yet to be fully understood, it has been shown that alpha and beta power at rest correlate positively with BOLD in the DMN and negatively with BOLD in the DAN (*Mantini et al., 2007*). Accordingly, increased activity in the DMN (red/yellow in state 1) and decreased activity in parietal regions of the DAN (blue in state 7) would both correspond to an increase in the BOLD signal. Based on this reasoning, we hypothesized that the antagonistic behavior of these two networks as shown in multiple fMRI studies would manifest in the inferred HMM states as an anticorrelation between the fractional occupancy time courses of these two states. In other words, periods of time in which the DMN state is frequently visited would coincide with periods of time in which the putative DAN state is rarely visited (and vice versa).

To this end, we computed the correlation coefficient between the fractional occupancy time courses for each state (computed within 10-s sliding windows as shown in *Figure 3E*). Positive correlations between a pair of states indicate that the two states are visited more frequently during similar periods of time. As predicted, we found strong antagonistic behavior between the DMN (state 1) and the putative DAN (state 7), suggesting an electrophysiological basis to the anticorrelated nature of these networks (*Figure 5A*).

We further assessed the relationship between different functional networks by examining the transitions between the inferred states. These transitions may be represented in the form of a transition matrix, where each row represents the probability of transitioning to any other state given the current state (*Figure 5B*). There is a clear structure to the matrix showing that transitions between certain pairs of states are more likely than others. A number of these transitions are intuitive, for example the strong probability of transitioning between visual states 2 and 6. There is a very low probability of transitioning between the DMN (state 1) and the putative DAN (state 7), raising the intriguing possibility that anticorrelation between these networks may arise from an inability of the system to transition directly between these two transient states.

## Occurrence of transient states reflect temporal variability in functional connectivity

A number of previous studies have assessed the temporal dynamics of resting state connectivity by computing measures of functional connectivity such as band-limited amplitude correlation within

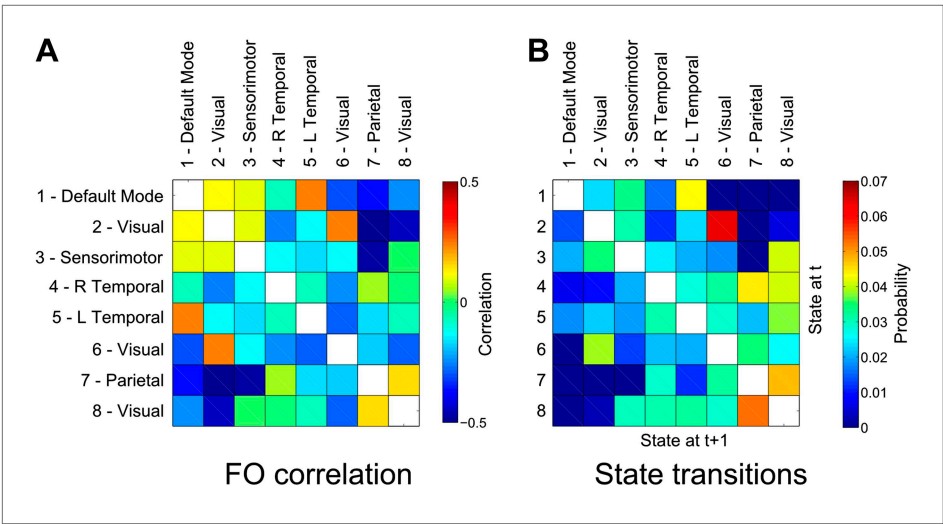

**Figure 5**. Relationship between states. (**A**) Correlation matrix between the fractional occupancy time courses of each state. Positive correlations between a pair of states indicate that the two states are visited more frequently during similar periods of time. (**B**) State transition matrix for the group HMM. The matrix shows the probabilities of transitioning to any particular state given the current state. The probability of remaining in the same state has been excluded from each matrix (shown in white).

sliding time windows of several seconds or longer (*Allen et al., 2012*; *Brookes et al., 2014*; *Chang and Glover, 2010*; *de Pasquale et al., 2012*). To investigate whether any relationship exists between temporal variability in amplitude correlation and the occurrence of states inferred from the HMM, we computed the sliding window envelope correlation between different nodes of the DMN and compared the resulting correlation time courses with changes in the fractional occupancy of the inferred DMN state. Regions of interest were defined in the centers of the nodes corresponding to the inferior parietal lobule (IPL), medial frontal gyrus (MFG) and medial temporal lobe (MTL) in both hemispheres (*Figure 6A*). The envelope of oscillatory activity was computed at these six locations for each subject. With reference to previous findings (*de Pasquale et al., 2010*), envelope correlation was computed within 10-s sliding windows between all six ipsilateral ROI pairs and all three contralateral ROI pairs. The HMM state corresponding to the DMN was identified and the fractional occupancy was computed within the same 10-s sliding window. The time courses of these two measures are shown for a single subject in *Figure 6B*, where the DMN pair shown is the right IPL and right MFG. In line with previous findings, it is evident that the envelope correlation between different network nodes alternates between periods of high and low correlation (*de Pasquale et al., 2010*). Interestingly, periods of high envelope correlation are generally associated with an increase in the fractional occupancy of the DMN state, suggesting that fluctuations in interregional functional connectivity represent periods of time in which a particular transient network is frequently visited. This relationship may be quantified as the correlation between the two time series for each subject and node pair. There is a clear positive correlation between the two measures for all inter- and intra-hemispheric pairs (*Figure 6C*).

## Discussion

We have characterized whole-brain spontaneous activity based on time-varying behavior of source localized MEG signals. Using a distinct methodology that is able to resolve changes in functional connectivity at high temporal resolution, we identified short-lived (100–200 ms) states that represent

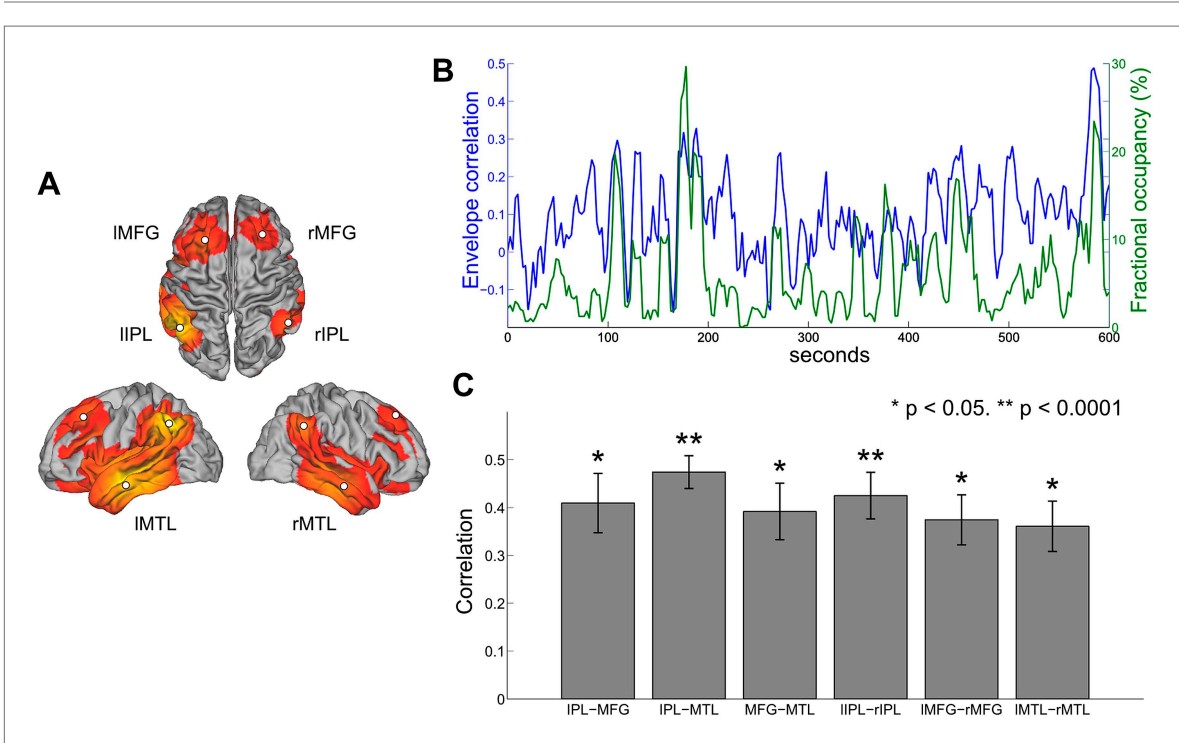

**Figure 6**. Comparison between HMM state occupancy and sliding window envelope correlation. (**A**) Six nodes identified from the DMN state (rIPL, lIPL, rMFG, lMFG, rMTG, lMTG). (**B**) Time courses of rIPL-rMFG envelope correlation (blue) and DMN state fractional occupancy (green) for a single subject computed using a 10-s sliding window (75% overlap between adjacent windows). (**C**) Correlation between the sliding window envelope correlation time course and fractional occupancy time course for each ipsilateral pair (bilaterally homologous pairs from the left and right hemispheres have been averaged together) and each contralateral pair (mean and s.e.m. over subjects).

unique spatiotemporal patterns of neural activity. The inferred states revealed spatial topographies that represent networks of spatially segregated brain regions that bear a strong similarity to seed based correlation or ICA derived RSNs in both MEG and fMRI. By assessing temporal changes in the occurrence of these states, we demonstrated that within-network functional connectivity is under-pinned by neuronal dynamics that fluctuate much more rapidly than has previously been shown.

## Large-scale transient networks of spontaneous oscillatory activity

By inferring an HMM from the amplitude envelopes of group-concatenated data, we identified spatial maps representing state specific increases in oscillatory amplitude in anatomical locations corresponding to well-known RSNs. These networks include the default mode, visual and sensorimotor networks and parietal regions of the DAN.

In addition to inferring the spatial covariance structure, the HMM also infers the time points at which each state is active. Each state was well represented over the group with all occupancies between 5% and 20% of the total time. The state life times were short, between 100 ms and 200 ms on average. These findings demonstrate that RSNs derived from conventional techniques such as ICA or seed–based correlation may be identified from discrete transient epochs that represent only a small fraction of the total recording, confirming recent studies from both fMRI and MEG (*Liu and Duyn, 2013*; *de Pasquale et al., 2012*).

Due to the assumption of mutual exclusivity of the HMM states, we should be cautious in automatically interpreting the fast switching between states in the form of short life times as a property of the underlying spontaneous activity. After all, an attractive hypothesis is that the brain's resting state activity consists of a finite number of unique networks that can overlap spatially and temporally (*Smith et al., 2012*). However, these relatively weak assumptions render the hypothetical networks unidentifiable using existing decomposition techniques (to the best of our knowledge), and so further constraints are currently needed to proceed. For example, in temporal ICA the assumption of temporal independence will discourage temporal overlap. In the HMM, the mutual exclusion approach will also prohibit temporal overlap. As a result, the HMM states can perhaps be best thought of as representing the most dominant unique configurations of these hypothetical networks. Based on this interpretation, the state time courses provide a meaningful window on the underlying network dynamics by indicating the most dominant state at each point in time. Indeed, we have shown that these time courses can provide an insight into the relationship between different functional networks (*Figure 5*).

Nonetheless, the rate of occurrence of the inferred states as described by the fractional occupancy allows us to assess the time scales at which within-network amplitude fluctuations are detectable. By modeling these fluctuations using the fractional occupancy time course computed at a range of different window widths, we show that they are best described by sub-second temporal dynamics with coordinated fluctuations on the order of 200–400 ms. Importantly, these time scales were not found for control surrogate data, where the fast network interactions were replaced by uncorrelated noise (see *Figure 4—figure supplement 1*), demonstrating that within-network functional connectivity is underpinned by neuronal dynamics that fluctuate much more rapidly than has previously been shown.

## Cross-network interactions

The HMM assumes that the states themselves are mutually exclusive. It might be thought that this assumption negates the possibility of the HMM providing insight into cross-network (i.e., across-state) interactions. While this is true at the fast (below 100 ms) within-state time scales, this does not mean that we cannot assess cross-network interactions at all. In particular, the HMM furnishes us with the probability of transitioning between different networks. This can tell us if there is a preference for the brain to move between two different networks, or an antagonism such that there is rarely a transition between two different networks. Indeed, the relationship between the DMN (state 1) and parietal areas of the DAN (state 7) was shown to be an example of the latter (*Figure 5B*). We have also demonstrated how cross-network interactions can be assessed using fractional occupancy time courses, which describe how frequently states are visited within 10-s time windows. By computing correlations between these slower time courses, we can assess relationships between networks at the time scales more typically associated with previous investigations into long-range resting state interactions (*Fox et al., 2005*). Of particular interest is the anticorrelation between the fractional occupancy time courses of the DMN state and the putative DAN state (*Figure 5A*).

The fact that cross-network relationships present in the fractional occupancy time courses are maintained when looking at the faster time scales of the state transitions, suggests a link between these two time scales. However, with the current methodology it is not possible to establish a causal link between these time scales (i.e., whether infrequent transitions arise because these networks are anticorrelated at longer time scales, or that this anticorrelation arises due to these infrequent transitions).

## Correspondence with BOLD RSNs

The states inferred by the HMM show some consistency with RSNs measured using fMRI. The spatial correspondence between the spatial maps obtained via the HMM and RSNs derived from application of ICA to BOLD are shown in *Figure 2—figure supplement 2*. A notable difference between these maps is the absence of the posterior cingulate cortex (PCC) or precuneus in the DMN state. One possible explanation is that the PCC may act as a functional 'hub', such that it is not strongly represented in any one state. While the HMM allows spatial overlap, we have visualized the states in terms of state-specific amplitude changes, such that any node that is active during a number of states will be suppressed in the spatial maps. It is possible that, as a hub, the PCC has membership in the majority of states and is therefore poorly identified by mapping state specific activity. However, it is not trivial to identify nodes that are active in multiple states because this is confounded by the spatial leakage due to the ambiguities in the source reconstruction (*Van Veen et al., 1997*; *Schoffelen and Gross, 2009*).

While there is strong evidence for hub-like behavior of the PCC/precuneus in terms of structural (*Sporns et al., 2007*; *Hagmann et al., 2008*) and functional (*Buckner et al., 2009*; *Tomasi and Volkow, 2011*) connectivity, evidence in MEG is more varied. Graph theoretical analyses of MEG band-limited power have revealed strong hubs in dorsal prefrontal cortex, lateral parietal cortex, and temporal cortex (in essence those areas represented by state 1), but notably not the PCC (*Hipp et al., 2012*). Furthermore, the PCC was not found to be present in RSNs derived from application of temporal ICA to MEG data (*Brookes et al., 2011*). Interestingly, an MEG-derived DMN comprising the PCC has been shown using seed–based correlation with the PCC as a seed, but only when restricted to time points in which the network was maximally correlated (*de Pasquale et al., 2012*).

Another reason why the PCC may not be present in the HMM is due to the relative insensitivity of MEG to deeper sources. This hypothesis is supported by the fact that in the present study, and in *Brookes et al. (2011)* and *Hipp et al. (2012)*, the sensor array comprised only axial gradiometers, which measure the spatial gradient of the field and are relatively insensitive to deep sources. Conversely, in *de Pasquale et al. (2012)*, the array comprised magnetometers that have increased depth sensitivity. It is therefore possible that the PCC is not present simply because it does not generate a measurable signal.

Finally, it is worth considering to what extent the DMN or DAN, and the particular spatial nodes they incorporate, are definitive networks. The particular form of such networks is tied to the approaches used to identify them, for example spatial ICA. Indeed when other approaches with different assumptions are made, then different networks can be inferred. One good example of this is that, when using temporal ICA on fMRI data, no single DMN is found (*Smith et al., 2012*). In short, the concept of a 'network' and the form they take depends on the assumptions made in the data decomposition approach, and none should be considered necessarily better than the other (assuming they show similar objective performance, e.g., Bayesian model evidence and reproducibility), but instead they each offer different perspectives on the nature of the brain's activity.

## Relation to EEG microstates

The short life times inferred from the HMM raises the question of whether there exists a relationship between the states shown here and the 'microstates' found from EEG studies (*Lehmann et al., 1998*; *Koenig et al., 2005*). Microstates are quasi-stable topographies in which the distribution of EEG power over the scalp remains stable for periods of around 100 ms. Clustering of these topographies into a limited number of classes has revealed that relatively few (typically four) unique maps are consistently identified across multiple time points and subjects (*Lehmann et al., 1998*). It has also been suggested that EEG microstates may represent the electrophysiological signatures of resting state networks. A number of recent studies have sought to investigate this by correlating fMRI RSNs with simultaneously acquired EEG recordings (*Britz et al., 2010*; *Musso et al., 2010*; *Yuan et al., 2012*).

Segmentation of EEG scalp maps into microstates is based on finding repeating distributions of power across multiple recording sites and therefore captures similar interactions to those that drive

the HMM. However, a clear distinction is that the HMM explicitly models the temporal dynamics and is therefore tuned to finding states that repeat in a predictable way. Another difference is that our approach exploits the superior spatial resolution available using MEG, by basing the inferred states on source space projections of MEG data. This makes results more directly interpretable in the context of fMRI resting state networks. While it is possible that the HMM states inferred in this paper relate to the source space counterparts of EEG microstates, without measuring EEG and MEG concurrently, a direct relationship cannot be confirmed.

## Relationship with functional connectivity at slower time scales

The states inferred using the HMM represent networks of activity that show some consistency with those found previously using spatial ICA on FMRI data (*Beckmann et al., 2005*; *Smith et al., 2009*) and using temporal ICA on MEG data (*Brookes et al., 2011*; *Luckhoo et al., 2012*). In both the ICA and HMM approaches, data from all voxels are used and therefore no prior spatial localization assumptions (e.g., seed voxels) are required. In the case of temporal ICA, two regions will tend to be strongly represented in the same network if their time courses exhibit a strong time-averaged correlation with the same independent component. By contrast, in the case of the HMM, membership of two regions to a particular state ('network') depends only on there being a repeated pattern of covariance at those points in time at which the state is active, that is, it is not time averaged over 'all' time points. The results in this paper, for example the fact that the two approaches result in similar networks, are consistent with the idea that ICA RSNs derive from the transient time-varying behavior captured by the HMM states. This idea is supported by the findings that seed-based functional connectivity is increased if evaluated only within those time points at which the rest of the network is synchronized (*de Pasquale et al., 2010*). In this paper, we provide evidence of a relationship between the sliding window correlation computed between nodes of the DMN and the occurrence rate of the DMN state (*Figure 6*). Accordingly, changes in within-network envelope correlation may reflect variations in the frequency at which a particular connectivity state is visited. One explanation for this observation is the idea that electrophysiological data are characterized by scale-free dynamics that span from hundreds of milliseconds to tens of seconds (*Van de Ville et al., 2010*). This fractal property of neural dynamics may provide an explanation for the similar spatial correlation structure that exists between signals at different temporal scales. Hence, slow fluctuations in band-limited amplitude correlations that underlie MEG RSNs may capture similar physiological phenomena as the HMM states, but seen through different temporal filters.

## Methodological considerations

In the present study, the data for individual subjects were temporally concatenated, yielding a single combined data set from which the HMM was inferred. Group concatenation is widely used in unsupervised analyses of resting state activity, particularly in the case of ICA (*Calhoun et al., 2009*). Nonetheless, it is important to bear in mind that this approach assumes that there is an anatomical correspondence between subjects. In the case of ICA, this assumption means that individuals share common group maps (*Calhoun et al., 2009*). In the case of the HMM approach, this assumption extends to the requirement that the data from different individuals within a particular state are drawn from the same multivariate normal distribution, and that the transitions between these states occur in a repeatable manner over subjects. This being said, there is no requirement that all states should be present in all subjects, however it is clear from the fractional occupancy time courses in *Figure 3E* that they are. In light of this assumption, states inferred from the group data may best be thought of as representing the spatiotemporal patterns of activity that occur most consistently over the group. An alternative strategy to group concatenation is to infer separate HMMs for each subject individually. This would allow the model to more freely adapt to individual subjects' patterns of activity and functional connectivity. However, a severe limitation of this approach is that there will not necessarily be a correspondence between states inferred from different subjects, making it difficult to perform subsequent analyses at the group level.

A limitation of the proposed technique is that HMM inference requires an a priori specification of the number of states, K. Bayesian inference techniques provide a means to test model order selection, by providing an approximation to the model evidence via the free energy. In theory, it should be possible to pick the optimal number of states by selecting the model with the greatest (negative) free energy. In practice however, we observe that the free energy increases monotonically up to K = 15 states, suggesting that the Bayes-optimal model may require an even higher number of states (*Figure 3— figure supplement 1A*). In the absence of a straightforward data driven approach to model order

selection, we opted instead to repeat the analysis for values of K from 4 to 15 and arbitrarily chose 8 states as the case to present here, which we believe represents a good trade-off between richness and redundancy. Results for different model orders are shown in *Figure 2—figure supplement 1*. Varying the number of states between 4 and 14 did not change the topographies of the most prominent RSN-like states. It is worth noting that a similar limitation exists for more established data-driven decompositions such as ICA, in which the choice of model order is driven by the application; for example, lower model orders are used to obtain the classic RSNs, and higher model orders are used to obtain finer grained parcellations for use in subsequent network analysis (*Smith et al., 2011*).

Finally, the observation model used in this paper corresponded to a multivariate normal distribution. The assumption of a Gaussian observation model allows inference of the HMM to be made tractable to variational Bayesian inference and thus permits its application to large amounts of data (40 observations and 1.5 hr of time points). However, it should be recognized that modeling only the first and second order statistics under a Gaussian assumption is likely to be an oversimplification of the underlying network dynamics. The multivariate normal distribution is just one of many potential observation models that may be used in the context of the HMM. For example, binary observation models have previously been used in the context of modeling interictal spikes (*Ossadtchi et al., 2005*). Future work will focus on implementing a multivariate autoregressive (MVAR) model that can model time lagged dependencies between observations.

### Functional role of transient synchronization

The work presented here rests on the underlying assumption that resting state activity may be broken down into a set of distinct connectivity patterns that repeat over time and where only one functional state may be active at any one time. In other words, the states inferred by the HMM are mutually exclusive. While this assumption may be an oversimplification of the underlying network dynamics, the concept that there exist distinct functional connectivity states that recur at different points in time is compatible with computational models of neuronal connectivity (*Deco et al., 2011*) and observations from both fMRI (*Allen et al., 2012*) and EEG (*Britz et al., 2010*; *Musso et al., 2010*; *Yuan et al., 2012*). The idea that RSNs represent states in which distributed cortical areas synchronize transiently is also compatible with the idea of a 'dynamic repertoire' of states that are continuously explored in order to more quickly adopt the network configuration optimal for a given impending input (*Deco et al., 2011*). This organization of dynamic activity through transient spatial patterns of coordination may provide the flexibility required to adapt to the rapidly changing computational demands of cognitive processing (*Bressler and Tognoli, 2006*).

## Materials and methods

### Data acquisition

Resting state MEG data were acquired from nine healthy subjects. The subjects were asked to lie in the scanner with their eyes open while 10 min of data were recorded. The MEG data were acquired using a 275 channel CTF whole-head system (MISL, Conquitlam, Canada) at a sampling rate of 600 Hz with a 150 Hz low pass anti-aliasing filter. Synthetic third order gradiometer correction was applied to reduce external interference. Localization of the head within the MEG helmet was achieved using three electromagnetic head position indicator (HPI) coils. By periodically energizing these coils their position within the MEG sensor array was identified. Prior to data acquisition, the HPI coil locations, the position of three fiducial points (the nasion, and left and right preauricular points), and the head shape were recorded using a three-dimensional digitizer (Polhemus Isotrack). MR images were acquired using a 3T Phillips Achieva MR system at $1 \times 1 \times 1$ mm$^3$ resolution running an MPRAGE sequence. Each subject's structural MRI was registered to the MNI152 standard brain such that all subsequent source space analysis was performed in MNI space. The locations of the MEG sensors with respect to the anatomy were determined by registering the digitized head surface to the head surface extracted from the structural MRI.

### Data pre-processing

The data were converted to SPM8 http://www.fil.ion.ucl.ac.uk/spm and down sampled to 200 Hz. Each recording was visually inspected to identify channels and/or periods of data containing obvious artifacts or with abnormally high variance, which were discarded. Independent component analysis (ICA) was used to decompose the sensor data for each session into 150 temporally independent components (tICs) and associated sensor topographies (http://research.ics.aalto.fi/ica/fastica). Artifact components were

classified via the following procedure. Eye-blink, cardiac and mains interference components were manually identified by the combined inspection of the spatial topography, time course, kurtosis of the time course and frequency spectrum for all components. Eye-blink artifacts typically exhibited high kurtosis (>20), a repeated blink structure in the time course and very structured spatial topographies. Cardiac component time courses strongly resembled the typical ECG signals, as well as having high kurtosis (>20). Mains interference had extremely low kurtosis (typically <−1) and a frequency spectrum dominated by 50 Hz line noise. Following artifact rejection the data were frequency filtered into the 4–30 Hz band.

## Source analysis

The pre-processed MEG data were projected onto a regular 8-mm grid spanning the entire brain using a custom scalar beamformer implemented in SPM8 (*Van Veen et al., 1997*; *Vrba and Robinson, 2001*; *Woolrich et al., 2011*). Beamforming is an adaptive spatial filter in which the estimated neuronal electrical activity $q_{r,\varphi}(t)$ at a pre-determined brain space location $r$ with orientation $\varphi$, and at time $t$ is given by a weighted sum of the $N$ sensor measurements

$$q_{r,\varphi}(t) = w_{r,\varphi}^{\mathsf{T}} m(t)$$

where, $w_{r,\varphi}$ is a ($N \times 1$) set of weights that govern the projection of the ($N \times 1$) sensor data $m(t)$ into source space, and superscript T indicates a transpose. Here, we use a linearly constrained minimum variance (LCMV) scalar beamformer (*Van Veen et al., 1997*; *Robinson and Vrba, 1999*; *Woolrich et al., 2011*). The weights are determined by an estimate of the ($N \times N$) covariance matrix $C$ computed between all sensor pairs, and a set of lead fields $h_{r,\varphi}$ that describes how a unit current generated by a dipolar source at location $r$ with orientation $\varphi$ would be measured at each MEG sensor (*Sarvas, 1987*; *Huang et al., 1999*)

$$w_{r,\varphi}^{\mathsf{T}} = \left[ h_{r,\varphi}^{\mathsf{T}} C^{-1} h_{r,\varphi} \right]^{-1} h_{r,\varphi}^{\mathsf{T}} C^{-1}$$

where, $h_{r,\varphi}$ is the ($N \times 1$) projection of the ($N \times 3$) lead fields $H_{r,\varphi}$ to the dipole orientation $\varphi$ that maximizes the projected signal-to-noise ratio, computed as in *Sekihara et al. (2001)*. The sensitivity of the beamformer varies for different locations in the head (*Van Veen et al., 1997*; *Vrba and Robinson, 2001*). To account for this spatial bias, the projected data $q_{r,\varphi}(t)$ were scaled by an estimate of the projected noise:

$$z_{r,\varphi}(t) = \frac{q_{r,\varphi}(t)}{w_{r,\varphi}^{\mathsf{T}} w_{r,\varphi}}$$

where, $w_{r,\varphi}^{\mathsf{T}} \times w_{r,\varphi}$ represents the projection of uncorrelated sensor noise (i.e., the noise covariance matrix is the identity matrix).

## Computation of source space amplitude envelopes

Following beamformer projection, the oscillatory amplitude envelope at each voxel was derived by computing the magnitude of the Hilbert transform of the source-reconstructed data. For computational efficiency, the envelopes were down sampled to 40 Hz by temporally averaging within sliding windows with a width of 100 ms and 75% overlap between consecutive windows. The amplitude envelopes were concatenated temporally across all subjects after spatially smoothing with a Gaussian kernel (FWHM 9.4 mm). The envelope data for each subject were demeaned and normalized by the global (over all voxels) variance prior to concatenation.

## Hidden Markov model

The group-concatenated envelopes were demeaned and pre-whitened to reduce the data to 40 principal (temporal) components with unit variance and zero mean. An HMM with 8 states was inferred (although see *Figure 2—figure supplement 1* and *Figure 3—figure supplement 1* for equivalent results with other model orders). Each inferred HMM state is associated with a unique multivariate normal distribution over the observations (principal components), defined by a ($M \times 1$) mean vector and a ($M \times M$) covariance matrix where $M = 40$ (the number of principal components) (*Rezek and Roberts, 2005*; *Woolrich et al., 2013*). To account for variations in the inference due to different initializations, 10 realizations were performed for each inference and the model with the lowest free energy was chosen. The most probable a posteriori state $u_t$ at each time point and was obtained using

the Viterbi algorithm (**Rezek and Roberts, 2005**). State time courses were defined for each state as indicator variables that indicate the points in time in which that state is most probable $u_t == k$. The HMM toolbox and example scripts may be downloaded from www.fmrib.ox.ac.uk/~woolrich/HMMtoolbox.

## HMM inference

We assume an HMM of length $T$ samples, state space dimension $K$, hidden state variables $s=\{s_1...s_T\}$ and observed data $y=\{y_1...y_T\}$, where $y_t$ are the $(M \times 1)$ principal components at time $t$ computed from the group-concatenated envelope data. The full true posterior probability of the model is then given by:

$$P(y,s,\Theta) = P(s_0|\pi_0)\prod_t^T P(s_t|s_{t-1},\pi_t)P(y_t|s_t,\theta)P(\pi_t)P(\pi_0)P(\theta)$$

where, $P(\pi_t)$, $P(\pi_0)$ and $P(\theta)$ are chosen to be non-informative priors. The HMM parameters $\Theta=\{\pi_0,\pi_t,\theta\}$ consist of $\pi_0$ which parameterize the initial state probability $P(s_0)$, $\pi_t$ which determine the state transition probability $P(s_t|s_{t-1})$ and $\theta$ which describe the observation probabilities $P(y_t|s_t)$. Here, we assume that the probability to transition to another state depends only on the state that the system is in, and not on the path it took to get to its current state, that is, it is Markovian:

$$P(s_t|s_1...s_{t-1}) = P(s_t \mid s_{t-1}) = \pi_t$$

where, $\pi_t$ is the $(K \times K)$ transition probability matrix in which the element $(i,j)$ describes the probability of transitioning from state $i$ to state $j$ between time $t-1$ and time $t$.

The term $P(y_t|s_t,\theta)$, is the observation model. In this work, we assume that the observation model for state $k$ is a multivariate normal distribution with $\theta_k=\{\mu_k,\Sigma_k\}$, where $\mu_k$ is the $(M \times 1)$ mean vector, and $\Sigma_k$ is the $(M \times M)$ covariance matrix:

$$P(y_t|s_t = k,\theta) \sim \mathcal{N}(\mu_k,\Sigma_k)$$

The prior distributions over the HMM parameters $\Theta=\{\pi_0,\pi_t,\theta\}$ are chosen to be conjugate distributions. The approximate posterior distributions will then be functionally identical to the prior distributions (i.e., a Gaussian prior density is mapped to a Gaussian posterior density), making the model tractable to certain kinds of inference. See **Rezek and Roberts (2005)** for details.

In this work, we use variational Bayes (VB) inference on the HMM, as described in **Rezek and Roberts (2005)**. This is fully probabilistic and furnishes us with full posterior distributions on the model parameters $P(\Theta,s,y)$. Aside from the inferred posterior distribution over the observation model $P(y_t|s_t,\theta)$, we are also interested in determining those points in time at which particular states are active. The relevant output is the marginal posterior inference on the state variables $P(s_t|y)$. This is obtained using Viterbi decoding. For the purpose of computing summary statistics of state life time and occupancy, we have chosen to hard classify the states as being on or off by choosing the most probable a posteriori state $u_t$ at each time point:

$$u_t = \arg \max_k P(s_t = k \mid y)$$

## Summary statistics

We defined a number of summary statistics to describe the temporal characteristics of the inferred states. For the purpose of computing these statistics, we have chosen to hard classify the states as being on or off by choosing the most probable a posteriori state $u_t$ at each time point:

*Fractional occupancy* is defined as the fraction of time spent in each state

$$fractional\,occupancy(k) = \frac{1}{T}\sum_t (u_t == k)$$

where, $u_t == k$ is one if $u_t = k$ and is zero otherwise, and $T$ is the length of the state sequence in samples. The *mean life time* is defined as the average amount of time spent in each state before transitioning out of that state:

$$mean\,life\,time\,(k) = \frac{\sum_t(u_t == k)}{number\,of\,occurences\,(k)}$$

The *mean interval length* is similarly defined as the average amount of time spent between consecutive visits to a particular state:

$$mean\ interval\ length\ (k) = \frac{T - \sum_t(u_t == k)}{number\ of\ occurences\ (k)}$$

where, the *number of occurrences* is given by:

$$number\ of\ occurrences(k) = \sum_t \left(\left((u_t == k) - (u_{t-1} == k)\right) == 1\right)$$

## Deriving partial correlation maps

By performing a principal component analysis prior to inferring the HMM, the dimensionality of the data was reduced to a computationally manageable amount. However, this means that the multivariate normal distributions that define each state span this reduced subspace, and are therefore not readily interpretable in terms of the underlying anatomy. To interrogate those brain areas associated with each state, we mapped state-specific activity by correlating the amplitude envelope at each voxel with the HMM state time courses. This has the advantage of identifying only activity that is unique to each state, and thus reduces components of the signal that are common across states that may obscure state-specific effects.

State specific changes in oscillatory activity were identified by computing the partial correlation of the state time courses with the source space data. The partial correlation was computed using a general linear modeling (GLM) framework as follows. The maximum a posteriori state $u_t$ at each time point was used to construct a ($K \times T$) design matrix $X$, where each column $k$ is an array of indicator variables that indicates whether state $k$ is on or off:

$$X(t,k) = \begin{cases} 1, & u_t = k \\ 0, & u_t \neq k \end{cases}$$

This design matrix, together with the full-rank (before whitening) source space data, was used in a GLM analysis (*Friston et al., 1996*; *Brookes et al., 2004*; *Woolrich et al., 2009*). Specifically, we perform a multiple linear regression at each voxel with the group-concatenated envelope data as the dependent variable. To compute the partial correlation, both the design matrix and the data were normalized to have zero mean and unit variance prior to fitting the GLM. This yields a set of $K$ spatial maps representing estimates of the partial correlation coefficient between each state and the data.

## Acknowledgements

The authors would like to thank Sofia Palazzo Corner for her assistance in acquiring the MEG data and Henry Luckhoo for his help with pre-processing the data. We thank the following institutions for their support: the RCUK Digital Economy program for provision of funding for APB through the Centre for Doctoral Training in Healthcare Innovation; the Leverhulme Trust for fellowship support for MJB and the University of Nottingham who funded the MEG scanner. MWW is funded by the Wellcome Trust, the MRC/EPSRC UK MEG Partnership award, and supported by the National Institute for Health Research (NIHR) Oxford Biomedical Research Centre based at Oxford University Hospitals Trust Oxford University (the views expressed are those of the author(s) and not necessarily those of the NHS, the NIHR or the Department of Health).

## Additional information

### Competing interests
TB: Reviewing editor, *eLife*. The other authors declare that no competing interests exist.

### Funding

| Funder | Grant reference number | Author |
| --- | --- | --- |
| Research Councils UK Digital Economy programme | | Adam P Baker |
| Wellcome Trust | | Mark Woolrich |

| Funder | Grant reference number | Author |
| --- | --- | --- |
| NIHR Oxford Biomedical Research Centre | | Mark Woolrich |
| Leverhulme Trust | | Matthew J Brookes |
| National Institutes of Health Human Connectome Project | 1U54MH091657-01 | Timothy Behrens |
| The Wellcome Trust | 088312AIA | Timothy Behrens |
| The Wellcome Trust | 098369/Z/12/Z | Stephen M Smith |
| UK MEG Partnership Award | MR/K005464/1 | Mark Woolrich |
| Engineering and Physical Sciences Research Council | EP/J012041/1 | Iead A Rezek |
| The Wellcome Trust and the Engineering and Physical Sciences Research Council | WT 088877/Z/09/Z | Penny J Probert Smith |

The funders had no role in study design, data collection and interpretation, or the decision to submit the work for publication. The views expressed are those of the authors and not necessarily those of the funders.

## Author contributions

APB, MW, Conception and design, Analysis and interpretation of data, Drafting or revising the article; MJB, Acquisition of data, Drafting or revising the article; IAR, SMS, TB, PJPS, Analysis and interpretation of data, Drafting or revising the article

## Ethics

Human subjects: The study was approved by the University of Nottingham Medical School Research Ethics Committee (approval code F/12/2006). All volunteers received a study information sheet, completed a safety questionnaire, and provided written informed consent, including consent to publish anonymised results.

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
