## [Decision Letter]

Thank you for sending your work entitled “Fast transient networks in spontaneous human brain activity” for consideration at *eLife.* Your article has been favorably evaluated by a Senior editor and 3 reviewers, one of whom, Jody Culham, is a member of our Board of Reviewing Editors.

The Reviewing editor and the reviewers discussed their comments before we reached this decision, and the Reviewing editor has assembled the following comments to help you prepare a revised submission.

All three reviewers described your manuscript as “interesting” and highlighted its novelty. One external reviewer stated, “Overall this is an excellent paper. The topic is important and timely. The results are provocative.”

While all three reviewers agreed that the paper will make a substantive contribution, there was also a consensus that the manuscript requires revisions to resolve several issues.

The following four main revisions are required (with further details from the reviewers appended for reference):

1) Discuss (or provide additional data to address) what the extracted states are reflecting in terms of network processing, particularly the degree to which they reflect stationarity and independence of networks.

2) Directly address the discrepancies between these results and fMRI RSC networks. Address the concern that some of the differences arise from the constraint that the networks not overlap (such that “hubs” like the precuneus aren't found).

3) Provide greater methodological detail and justification for the analytic choices made, particularly with respect to the rationale for choosing eight states and collapsing across subjects.

4) Do not overstate the links to EEG microstates.

*Point*
*1*

Reviewer 2:

Electrophysiological data are nonstationary, and periods of oscillatory activity are particularly so, as frequency and amplitude of neural oscillations are quite dynamic; in fact, nonstationarity can be regarded as a characteristic of oscillatory activity (Mäkinen, May, & Tiitinen, 2005). Since each extracted state was multivariate normal, the HMM technique as applied here might be detecting the most-stationary epochs in resting MEG data, which would suggest that rather than detecting networks, it is detecting epochs during which minimal oscillatory (i.e., network) activity is present in a given region. One simple means to assess this would be to describe the characteristic frequency of each of the 8 detected states using e.g., a Fourier transformation at the center of mass for each network as presented in Figure 2. If this would constitute too much additional work, the authors should provide a substantial and well-referenced limitations section. Given that HMM is also an effective method of spike detection in M/EEG data (Ossadtchi, Mosher, Sutherling, Greenblatt, & Leahy, 2005) the authors should describe in detail the methodological differences between HMM for spike detection and HMM as used in the present study.

Reviewer 3:

1. The method identifies states that are mutually exclusive. In order to study interactions between networks by definition the networks must be coupled. The authors propose to study interactions by analyzing the time course of the fractional occupancy. While this may be reasonable, the pros and cons of this approach, and how much traction we get biologically, must be clearly stated. At one extreme one could argue that given the methodology allows to tracks the temporal alternation of unique and independent states, it does not tell us anything about across–network interactions.

2. The identification of the brain states is driven using a fairly simple model. What is the justification for Gaussian models in building the posterior and non informative priors? Should there not be some prior validation on real data?

3. Relationship RSNs - brain states: please explain whether the approach is somehow circular: from MEG envelope - > PCA - > only 8 brain states and then again the authors compare the states with the data via correlation?

4. The HMM procedure to identify brain states exclude, by definition, cortical hubs, i.e., nodes involved in more systems (see for example the absence of PCC in DMN). How does this limit the interpretation of results?

Reviewer 3:

DMN-DAN anticorrelation. The transition between the states DMN-DAN does not relate to their anti-correlation in my view. It is just telling us that when the DMN is coupled the DAN is not and vice versa but this is obvious based on the HMM state extraction. This is true to some extent for all pairs of networks since again the states are mutually exclusive. The argument is based on the time course of the fractional occupancy (limitations noted above apply here).

Even if the nets were free to overlap, the results indicate that the transition between DMN and DAN are infrequent. This is in contrast with the fMRI literature where these two networks are strongly coupled, so should show correlated alternations according to this method. I think that the observed infrequent transitions are actually in favor of the opposite hypothesis, namely that these two networks are not correlated. Furthermore, the issue of anti-correlation between DMN and DAN should not be over-emphasized given that even in fMRI there are significant questions with regard to their biological significance.

*Point*
*2*

Reviewing Editor:

One keystone of the paper is the finding that at fine time scales as well as the coarser scale measured by fMRI, activation in the default mode network (DMN) is negatively correlated with activation in the dorsal attention network (DAN). However, the networks measured here show only partial overlap with the standard networks measured by fMRI. For example, the putative DMN network shown here does not include activation in the precuneus and the putative DAN only coarsely overlaps with the fMRI DAN. These discrepancies should be explicitly discussed.

Reviewer 3:

1. The RSN topographies do not seem standard even according to the authors' prior work:

a) DMN is missing posterior cingulate;

b) VIS is split in 3 separate states;

c) The parietal network is quite diffuse, and important nodes such as FEF are not seen? More importantly, what is the meaning of a state that is negatively correlated with the power envelope time course? Does the diffusiveness of the maps may suggest that these states are not truly within-network, but by themselves reflecting across network interactions? What is the actual spatial correlation between RSN measured on slow envelope and HHM states?

d) Previous work emphasizes bilateral networks, while here some networks are bilateral while other are more unilateral (e.g., temporal). Please comment.

2. If the brain states efficiently capture all dynamic states, one could expect that all of the brain is covered. What percentage of gray matter is covered by the network states? Are the holes, e.g., PCC potentially reflecting 'hub' like activity?

*Point*
*3*

Reviewer 2:

Significant methodological detail is omitted. No rationale is given for the number of states selected. It is unclear whether all states occurred in all subjects, and no rationale is given for temporal concatenation across subjects. As currently presented, it remains possible that the detected states are subject-specific. It is also unclear whether the authors used existing beamformer code (e.g., Fieldtrip; Oostenveld, Fries, Maris, & Schoffelen, 2011) or created their own. In addition, no details are given on possible interactions between beamforming and HMM. Finally, the authors state that after beamformer projection, downsampling, enveloping, and concatenation, the envelopes were pre-whitened into 40 principal components. Some information on the topography of these components should be provided.

Reviewer 2:

Some details are incorrect. For example, in their discussion of ICA in the section headed “Relationship with functional connectivity at slower time scales”, the authors state that “...two regions will tend to be strongly represented in the same component (“network”) if their time courses exhibit a strong time-averaged correlation over all time points.” This is true for spatial ICA as typically applied to fMRI data, but not for temporal ICA which would ordinarily be applied to MEG data.

*Point*
*4*

Reviewer 2:

The posited relationship to EEG microstates is conjecture, unsupported by the presented data. It is also unclear how infrequent bursts of neural activity underlying transient microstates would explain the sustained changes in blood oxygenation required to produce the BOLD timecourses observed in RSNs.

Reviewer 3:

The discussion points to a paper by Mantini et al 2007 showing that fMRI timeseries in DAN/DMN respectively were negatively/positively correlated with alpha/beta EEG power. However that relationship with EEG does not say anything about their mutual coupling.

---

## [Author Response]

We would like to thank the editor and the reviewers for their comments on the manuscript. We have responded to each of the reviewer points individually below.

Point 1

*Reviewer*
*2:*

*Electrophysiological data are nonstationary, and periods of oscillatory activity are particularly so, as frequency and amplitude of neural oscillations are quite dynamic; in fact, nonstationarity can be regarded as a characteristic of oscillatory activity (Mäkinen, May, & Tiitinen, 2005). Since each extracted state was multivariate normal, the HMM technique as applied here might be detecting the most-stationary epochs in resting MEG data, which would suggest that rather than detecting networks, it is detecting epochs during which minimal oscillatory (i.e., network) activity is present in a given region. One simple means to assess this would be to describe the characteristic frequency of each of the 8 detected states using e.g., a Fourier transformation at the center of mass for each network as presented in*
Figure 2*. If this would constitute too much additional work, the authors should provide a substantial and well-referenced limitations section*.

As the reviewer highlights, electrophysiological data, and oscillatory activity are characterized by highly non-stationary dynamics. Indeed, the principal motivation for our study was to assess resting state functional connectivity without assuming temporal stationarity across the duration of the recording. We also agree that when an HMM state is active this corresponds to short periods of time in which there is temporary stationarity – this is explicit in the model – and we can interpret the states as corresponding to the most repeatable patterns of oscillatory activity. However, the evidence in the paper suggests that these states do not correspond to when there is minimal oscillatory activity in the constituent regions. Instead, Figure 2 shows that most states/networks show an increase in the oscillatory power when the state is active compared to when it is inactive.

To make this clearer we have followed the reviewer’s suggestion, and produced power spectra, comparing the power of the oscillations between when each state is active and when it is inactive. This also provides a useful insight into the frequencies of the oscillations that underlie each network. We computed a time-frequency (TF) representation of the data at each voxel using Morlet wavelets (Morlet factor = 6) and assessed the power across frequency bands by averaging this TF representation both within state and outside of the state, using the time points indexed by the state time courses. We computed a weighted average spectrum by weighting each voxel’s power spectrum using the spatial maps in Figure 2. This ensures that the spectra represent the entire network, rather than a single node within the network. It is clear that states whose spatial maps show positive correlations with the state time courses represent increases in oscillatory activity. There is also evidence for some frequency specificity.Author response image 1.Power spectra for each state.The (pre-enveloped) data for each subject and at each voxel were decomposed into a time-frequency representation using Morlet wavelets (Morlet factor = 6). The magnitude of the transformed data was averaged both inside and outside of the state, using the time points indexed by the state time courses, resulting in an inside-state and outside-state power spectrum at each voxel. Weighted averages of these spectra were computed using the spatial maps in Figure 2. The red and blue plots show the averaged power spectra computed inside and outside the state respectively. The shaded region shows the s.e.m. over all subjects and the circles show the frequencies corresponding to each wavelet scale used.

*Given that HMM is also an effective method of spike detection in M/EEG data (Ossadtchi, Mosher, Sutherling, Greenblatt, & Leahy, 2005) the authors should describe in detail the methodological differences between HMM for spike detection and HMM as used in the present study*.

We have added a reference to (Ossadtchi et al., 2005) and added the following to the discussion: “The multivariate normal distribution is just one of many potential observation models that may be used in the context of the HMM. For example, binary observation models have previously been used in the context of modeling interictal spikes (Ossadtchi et al., 2005).”

*Reviewer*
*3:*

*1. The method identifies states that are mutually exclusive. In order to study interactions between networks by definition the networks must be coupled. The authors propose to study interactions by analyzing the time course of the fractional occupancy. While this may be reasonable, the pros and cons of this approach, and how much traction we get biologically, must be clearly stated. At one extreme one could argue that given the methodology allows to tracks the temporal alternation of unique and independent states, it does not tell us anything about across-network interactions*.

The states themselves are by definition mutually exclusive. This strong assumption has the benefit of making the model robust and identifiable. However, as with any assumption in a data-driven decomposition, we agree with the reviewer that this has an impact on the interpretation. To clarify the impact of this assumption we have expanded the part of the Discussion that was addressing this.

*2. The identification of the brain states is driven using a fairly simple model. What is the justification for Gaussian models in building the posterior and non informative priors? Should not there be some prior validation on real*
*data?*

We are assuming that the assumption of Gaussianity (within state) is a sufficiently good first order approximation. Indeed, the results show that this is the case, to the extent that the results are biologically meaningful and plausible. However, we acknowledge that it is also possible that other HMM observation models could be better. The principled way to assess different observation model assumption is to do so within the context of the HMM model, using measures such as the Bayesian model evidence, or cross validation. While future work will focus on implementing more advanced observation models, this extension is outside of the scope of the current paper. We have clarified this by expanding the relevant part in the Discussion.

*3. Relationship RSNs - brain states: please explain whether the approach is somehow circular: from MEG envelope - > PCA - > only 8 brain states and then again the authors compare the states with the data via*
*correlation?*

The approach used to obtain state-specific spatial maps is not circular. There is no constraint in the HMM inference (e.g., there are no pre-specified seeds or spatial map priors) that could bias the form of the HMM inference in this way. The correlation of the state time courses back onto the data is simply a way of projecting the states from the reduced subspace to the original voxel space. An alternative would be to project the observation model means and covariance matrices using the eigenvectors derived from the original PCA. However, performing a multiple regression (partial correlation) has the advantage of identifying only activity that is unique to each state, and thus reduces components of the signal that are common across states, which may otherwise obscure state-specific effects. To make this clearer we have added to the Materials and Methods section the paragraph beginning: “By performing a principal component analysis prior to inferring the HMM, the dimensionality of the data was reduced to a computationally manageable amount.”

*4. The HMM procedure to identify brain states exclude, by definition, cortical hubs, i.e., nodes involved in more systems (see for example the absence of PCC in DMN). How does this limit the interpretation of*
*results?*

While the states inferred by the HMM are temporally mutually exclusive, they do allow overlap spatially. The spatial structure is encoded via the observation model (multivariate normal distribution). For example, nodes A and B may be active during state 1, but nodes A and C during state 2. These different patterns will be represented by an increase in the mean and/or variance of the state’s MVN. So in this regard there is no reason why a hub, such as the PCC, should not be represented. However, it is possible that hub activity may be suppressed by the particular method we use for spatially mapping state-specific activity. The mapping uses partial correlation of the state time course with the amplitude envelope to identify brain areas whose band-limited amplitude increases or decreases specifically within a particular state. However, a hub node, through its inclusion in multiple states, might not show state specific increases or decreases in amplitude, and so will not be represented within any one state’s spatial map. We have added to the discussion in the context of the PCC’s absence to make this clearer.

*Reviewer*
*3:*

*DMN-DAN anticorrelation. The transition between the states DMN-DAN does not relate to their anti-correlation in my view. It is just telling us that when the DMN is coupled the DAN is not and vice versa but this is obvious based on the HMM state extraction. This is true to some extent for all pairs of networks since again the states are mutually exclusive. The argument is based on the time course of the fractional occupancy (limitations noted above*
*apply here)*.

*Even if the nets were free to overlap, the results indicate that the transition between DMN and DAN are infrequent. This is in contrast with the fMRI literature where these two networks are strongly coupled, so should show correlated alternations according to this method. I think that the observed infrequent transitions are actually in favor of the opposite hypothesis, namely that these two networks are not correlated. Furthermore, the issue of anti-correlation between DMN and DAN should not be over-emphasized given that even in fMRI there are significant questions with regard to their biological significance*.

We appreciate that any statements about correlation need to indicate clearly which particular time courses (from each state) are used to compute the correlation. It is the slower fractional occupancy (FO) time courses that are empirically anti-correlated between the DMN and parietal areas of the DAN. The fact that these FO time courses are anticorrelated, and that the FOs are derived from the HMM state time courses, suggests that these FO anti-correlations (and potentially associated BOLD RSN anti-correlations at the same time scale) are related to the antagonistic behavior identified at the faster time scales of the state time courses. To make this clear we have ensured that any statements about correlation indicate clearly which time courses are being correlated (e.g., FO), and we have also added to the Discussion the paragraph beginning: “The fact that cross-network relationships present in the fractional occupancy time courses are maintained …” We address the issue of the “biological significance” of the DAN and DMN in a later response to a reviewer’s point.

Point 2

*Reviewing*
*Editor:*

*One keystone of the paper is the finding that at fine time scales as well as the coarser scale measured by fMRI, activation in the default mode network (DMN) is negatively correlated with activation in the dorsal attention network (DAN). However, the networks measured here show only partial overlap with the standard networks measured by fMRI. For example, the putative DMN network shown here does not include activation in the precuneus and the putative DAN only coarsely overlaps with the fMRI DAN. These discrepancies should be explicitly discussed*.

We have added an analysis of the spatial correspondence of the inferred HMM spatial maps to the classic fMRI spatial ICA derived RSNs. This shows that there is generally good correspondence. See Figure 2—figure supplement 2. However, as pointed out by the reviewing editor, we also acknowledge that there is a degree of dissimilarity between the networks obtained via the HMM and those measured by fMRI (although this discrepancy is less so with those identified in MEG using temporal ICA – as in (Brookes et al., 2011)).

We have addressed each of the points on this topic in the response to the specific reviewer’s comments below. However, generally we now more clearly acknowledge that we do not necessarily expect complete correspondence between networks identified in this work, and those previously identified using spatial ICA on fMRI data. In particular, to make this clear we have added to the Discussion the paragraph beginning: “Finally, it is worth considering to what extent the DMN or DAN, and the particular spatial nodes they incorporate, are definitive networks.”

*Reviewer*
*3:*

*1. The RSN topographies do not seem standard even according*
*to the authors' prior work:*

*a) DMN is missing posterior*
*cingulate;*

As well as the response to the general point, we have added three paragraphs to the Discussion to address the specific lack of the PCC, beginning: “One possible explanation is that the PCC may act as a functional “hub”, such that it is not strongly represented in any one state.”

*b) VIS is split in 3 separate*
*states;*

Multiple visual networks have also been found in the case in BOLD RSNs. For instance, in (Smith et al., 2012), distinct ICA components were found representing medial, occipital pole, and lateral visual areas.

*c) The parietal network is quite diffuse, and important nodes such*
*as FEF are not seen?*

We agree that the parietal network is diffuse and have moderated our claim that it unambiguously represents the dorsal attention network. We now refer to this network as “parietal regions of the DAN” or the “putative DAN” when discussed in terms of its relationship with the DMN. Nonetheless, the fact that it is strongly anticorrelated with the DMN adds evidence that this state encompasses at least some aspects of the DAN network. As previously discussed, we must also consider that the use of both MEG and the HMM will result in a decomposition that is somewhat distinct from that of BOLD ICA.

*More importantly, what is the meaning of a state that is negatively correlated with the power envelope time*
*course?*

A state that correlates negatively with the power envelope time course indicates that band-limited MEG power decreases at those time points in which the state is active. This is now made clearer in the Methods section as follows: “Each map represents the partial correlation between the state time courses and the group-concatenated amplitude envelope at each voxel. Accordingly, state specific increases and decreases in amplitude are represented by red/yellow and blue colors respectively.”

*Does the diffusiveness of the maps may suggest that these states are not truly within-network, but by themselves reflecting across network*
*interactions?*

We suggest that the diffusivity of the HMM spatial maps in Figure 2 as compared with fMRI, arises due to the reduced spatial resolution of beamformed MEG. Indeed the spatial specificity is comparable to the spatial maps found using temporal ICA in (Brookes et al., 2011). The apparent extra diffusivity of states 6-8 in Figure 2 is due to the use of the same percentile threshold for all 8 states, combined with states 6-8 being less strong.

*What is the actual spatial correlation between RSN measured on slow*
*envelope and HHM states?*

As mentioned earlier, we have now included Figure 2—figure supplement 2 to allow a more direct comparison of the inferred states with BOLD RSNs and have stated the spatial correlation in this figure.

*d) Previous work emphasizes bilateral networks, while here some networks are bilateral while other are more unilateral (e.g., temporal). Please*
*comment.*

While a large proportion of previously published RSNs are bilateral, a significant minority are unilateral. These observations are consistent across BOLD RSNs (e.g., Smith et al., 2009) and MEG RSNs (Brookes et al., 2011).

*2. If the brain states efficiently capture all dynamic states, one could expect that all of the brain is covered. What percentage of gray matter is covered by the network states? Are the holes, e.g., PCC potentially reflecting 'hub' like*
*activity?*

It is true that, taken together, the states should explain the mean and variance across the entire brain. However, because the spatial maps shown in Figure 2 represent state specific activity (i.e., increases or decreases in amplitude that correlate with each state time course) regions whose activity is common across states will be suppressed, resulting in the apparent “holes”. Another possibility is that the PCA used to reduce the dimensionality of the data removes these regions. To address these issues directly we have now included spatial maps of the variance before and after the PCA, and an averaged map of all the states to demonstrate the proportion of gray matter covered by the network states (Figure 8). As mentioned earlier we have included a discussion of hubs, and specifically the PCC, in the revised manuscript.Author response image 2.Spatial maps showing (**A**) pre- and (**B**) post- PCA variance of the 4-30 Hz band limited envelopes averaged over subjects. The maps in (**C**) show the average of the unthresholded states’ spatial maps (the sign was ignored when taking the average, to avoid state-specific increases and decreases from cancelling each other out).

Point 3

*Reviewer*
*2:*

*Significant methodological detail is omitted. No rationale is given for the number of states selected. It is unclear whether all states occurred in all subjects, and no rationale is given for temporal concatenation across*
*subjects.*

We have included a section in the discussion titled “Methodological considerations” and have specifically clarified some of the methodological details and rationale. In terms of the choice of the number of inferred states, we have added the paragraph beginning: “A limitation of the proposed technique is that HMM inference requires an a priori specification of the number of states, K. Bayesian inference techniques provide a means to test model order selection, by providing an approximation to the free energy.”

In terms of the choice of the use of temporal concatenation across subjects, we have added the paragraph beginning: “Group concatenation is widely used in unsupervised analyses of resting state activity, particularly in the case of ICA (Calhoun et al., 2009).”

*“It is unclear whether all states occurred in all subjects” and “As currently presented, it remains possible that the detected states are*
*subject-specific.”*

The fractional occupancy time courses in Figure 3 show that this is not the case. We have added the following to the Results section to make this clearer: “These fractional occupancy time courses reveal slower temporal changes in the occurrence of the HMM states. These time courses are shown for all subjects in Figure 3. It is clear that each state was represented in all subjects.”

*It is also unclear whether the authors used existing beamformer code (e.g., Fieldtrip; Oostenveld, Fries, Maris, & Schoffelen, 2011) or created their*
*own*.

We used a custom beamformer adapted from the one in SPM8 and based on the standard scalar LCMV beamformer (Van Veen et al., 1997). We have amended the following text to make this clear: “The pre-processed MEG data were projected onto a regular 8 mm grid spanning the entire brain using a custom scalar beamformer implemented in SPM8.”

*In addition, no details are given on possible interactions*
*between beamforming and HMM*.

Beamformer source localization has been shown to induce spurious correlations due to “signal leakage” between voxels with correlated spatial filter weights. However, the covariance due to signal leakage will be stationary over time, whereas the HMM states reflect covariance that is unique to each state. As a result, inference of the HMM states is unlikely to be driven by signal leakage.

Furthermore, we inferred an HMM for a single subject in both source and sensor space, and used the inferred state time courses to assess state specific amplitude fluctuations across the brain (using the source space data). The two approaches yielded similar spatial maps, demonstrating that the beamforming process does not affect the inferred states (Figure 9).Author response image 3.Spatial maps showing the partial correlation of the source space amplitude envelopes with state time courses of an HMM inferred in source space and sensor space.Both approaches reveal very similar maps for the default mode network (top), visual network (middle) and sensorimotor network (bottom).

*Finally, the authors state that after beamformer projection, downsampling, enveloping, and concatenation, the envelopes were pre-whitened into 40 principal components. Some information on the topography of these components should be*
*provided.*

We have included the spatial topography of all 40 principal components in Figure 10Author response image 4.Spatial topographies of the 40 principal components, arranged in order of their variance (top to bottom, left to right).The cross hairs are positioned at the center of the head (MNI coordinates [0 -18 18]).

Reviewer 2:

*Some details are incorrect. For example, in their discussion of ICA in the section headed “Relationship with functional connectivity at slower time scales”, the authors state that “...two regions will tend to be strongly represented in the same component (“network”) if their time courses exhibit a strong time-averaged correlation over all time points.” This is true for spatial ICA as typically applied to fMRI data, but not for temporal ICA which would ordinarily be applied to MEG data*.

We thank you for pointing this out. In the case of temporal ICA, regions will be represented in the same network if they share a time-averaged correlation with the independent time course, though not necessarily with each other. We have rephrased the sentence to now read: “In the case of temporal ICA, two regions will tend to be strongly represented in the same network if their time courses exhibit a strong time-averaged correlation with the same independent component.”

Point 4

*Reviewer*
*2:*

*The posited relationship to EEG microstates is conjecture, unsupported by the presented data. It is also unclear how infrequent bursts of neural activity underlying transient microstates would explain the sustained changes in blood oxygenation required to produce the BOLD timecourses observed in RSNs*.

We acknowledge that the relationship to EEG microstates is speculation and that without measuring EEG and MEG concurrently a direct relationship between the HMM states and microstates cannot be confirmed. We have shortened our discussion of EEG microstates accordingly and have added the following: “While it is possible that the HMM states inferred in this paper relate to the source space counterparts of EEG microstates, without measuring EEG and MEG concurrently, a direct relationship cannot be confirmed.”

*Reviewer*
*3:*

*The discussion points to a paper by Mantini et al 2007 showing that fMRI timeseries in DAN/DMN respectively were negatively/positively correlated with alpha/beta EEG power. However that relationship with EEG does not say anything about their mutual coupling*.

We refer to the Mantini et al., 2007 paper as it provides an explanation for the fact that amplitude increases within the DMN state correlate negatively with amplitude decreases within the putative DAN state. A key finding of Mantini et al., 2007 was that EEG band-limited power has an inverse relationship with BOLD within the DAN. This suggests that the cross-network interactions observed in BOLD RSNs and the MEG HMM states are mutually consistent. We acknowledge that the paper by Mantini et al., 2007 does not provide the exploration of the coupling between DMN and DAN band-limited power that would be necessary to more strongly support or refute our claim.